# Sunpheno: A Deep Neural Network for Phenological Classification of Sunflower Images

**DOI:** 10.3390/plants13141998

**Published:** 2024-07-22

**Authors:** Sofia A. Bengoa Luoni, Riccardo Ricci, Melanie A. Corzo, Genc Hoxha, Farid Melgani, Paula Fernandez

**Affiliations:** 1Laboratory of Genetics, Wageningen University & Research, 6708 PB Wageningen, The Netherlands; sofia.bengoaluoni@wur.nl; 2Department of Information Engineering and Computer Science, University of Trento, 38123 Trento, Italy; ricardo.ricci@unitn.it (R.R.); farid.melgani@unitn.it (F.M.); 3IABIMO UEDD INTA CONICET, Buenos Aires 1686, Argentina; corzo.melanie@inta.gob.ar; 4Faculty of Electrical Engineering and Computer Science, Technische Universität Berlin, 10587 Berlin, Germany; genc.hoxha@unitn.it

**Keywords:** phenology, senescence, sunflower, deep machine learning

## Abstract

Leaf senescence is a complex trait which becomes crucial for grain filling because photoassimilates are translocated to the seeds. Therefore, a correct sync between leaf senescence and phenological stages is necessary to obtain increasing yields. In this study, we evaluated the performance of five deep machine-learning methods for the evaluation of the phenological stages of sunflowers using images taken with cell phones in the field. From the analysis, we found that the method based on the pre-trained network resnet50 outperformed the other methods, both in terms of accuracy and velocity. Finally, the model generated, Sunpheno, was used to evaluate the phenological stages of two contrasting lines, B481_6 and R453, during senescence. We observed clear differences in phenological stages, confirming the results obtained in previous studies. A database with 5000 images was generated and was classified by an expert. This is important to end the subjectivity involved in decision making regarding the progression of this trait in the field and could be correlated with performance and senescence parameters that are highly associated with yield increase.

## 1. Introduction

Sunflower is a primary oil crop, contributing to 2% of the total harvest area worldwide and with a global gross production value of 21.4 billion USD in 2020 [1]. Despite their economic importance, sunflower breeding has shown a low increase in yield in comparison with wheat, rice, soybean, and maize [1], with its trait of early leaf senescence being one of the most relevant in terms of the gap between real and potential yields for this crop [2,3,4].

Leaf senescence is the last stage of leaf development and is characterized by an active decline in the photosynthetic rate, rupture of chloroplasts, and cell death. In annual plants, such as grain and oil crops, flowering induces senescence accompanied by nutrient remobilization (mainly nitrogen) from leaves to developing seeds [5,6,7]. The remobilization of nutrients must be coordinated with the developmental stages of the sunflower. If the senescence is anticipated to occur at flowering time or if the rate of senescence increases disproportionately during the development of the seeds, losses in yield are observed [2,4,8,9]. The plant’s ability to maintain an active photosynthetic area for longer periods might improve the recycling process and, consequently, the grain content [9].

Crop phenological information is not only important in relation to senescence, but it is also used in yield modeling and crop monitoring. The impact of climatic conditions on final crop yield depends on the phenological stage [10,11,12]. In sunflower, a reduction in the intercepted radiance by shading has a different impact on the grain number according to the developmental stage in which the shading was applied. Meanwhile, the grain set of florets in the central position on the head radius is affected when shadowing takes place in early or late post-anthesis periods, while the grain set of florets in the mid-section was only affected by early anthesis shading. Thus, the post-anthesis period is critical for determining grain number [13]. On the other hand, the reproductive period of pre-anthesis has an impact on the determination of grain size, a parameter also correlated with the maximum seed weight [14]. Finally, the early post-anthesis period is highly susceptible to intercepted radiation, being decisive for the weight and oil content of the grains [15,16].

In sunflower, the phenological stages were described by Schneiter and J.F. Miller in 1981 [17]. According to this classification, the vegetative stages are named as ‘V’ following the number of fully expanded leaves, and the reproductive stages are named from R1 (the first appearance of the inflorescence) to R9 (physiological maturity). Furthermore, stage R5 can be divided into substages R5.1 to R5.9, which describe the percentage of anthesis in the head area. The evaluation of crop phenology is conventionally identified by monitoring activities in the field. These activities provide an accurate description of crop phenology, but are also expensive and time consuming, and therefore rarely implemented on a large scale.

New technologies for describing morphophysiological parameters in sunflower fields have been developed over the past decade [18,19,20]. The use of new types of sensors in the field and their combination with mathematical models, such as deep machine learning algorithms, accelerates the evaluation time, homologates the measurement between different work groups, and improves the accuracy and resolution [21,22]. Automatic evaluation allows farmers and experts to decrease the cost of analysis by orders of magnitude, making it possible to obtain a denser temporal analysis of plant phenological stages in the field. 

Neural networks (NNs) are mathematical models consisting of a series of layers formed by neurons. Briefly, we can classify NNs in three different classes: fully connected feedforward NNs (also called multi-layer perceptron (MLP)), convolutional NNs (CNNs) and recurrent NNs (RNNs). Convolutional neural networks (CNNs) have shown remarkable success in computer vision tasks such as image classification, object detection, and segmentation. CNNs are inspired by the structure of the visual cortex in animals, which is responsible for processing visual information. The key idea behind CNNs is to learn a hierarchical representation of the input image by convolving filters over the image and pooling the results to produce a feature map. This allows the network to capture local patterns in the image, such as edges and textures, and gradually build up to more abstract representations of objects and scenes. Researchers have discovered that, thanks to their capability to build a gradual understanding of objects by stacking simpler representations, convolutional neural networks are also very suitable for transfer learning. Transfer learning enables practitioners to transfer the knowledge obtained on a dataset for a specific task, to a new dataset with a possibly similar but not equivalent task.

In the context of image classification, several pretrained architectures have been introduced, mainly trained on ImageNet [23] for image classification. We will experiment with both non pretrained and pretrained architectures, including resnet18, resnet50, vgg16 and a pretrained vision transformer [24,25].Transformers are a different family of deep learning algorithms, based entirely on the attention mechanism. They have been introduced in text translation, and were shortly after adopted for other natural language processing (NLP) tasks, such as sentiment classification, topic identification [26,27,28,29] and text generation. Transformers have also been adapted in the computer vision field and applied for the first time for image classification, showing promising results. 

In this study, five deep machine-learning methods, including CNNs and transformers, were evaluated for the automatic discrimination of sunflower developmental stages. The best model was then used to evaluate the progression of phenological stages in lines R453 and B481_6 on images acquired with three smartphones. These two inbred lines belonging to the INTA Sunflower Breeding Program and the INTA Manfredi Sunflower Germplasm Collection, have been described previously by the group as contrasting senescence phenotypes that also present a difference in yield. 

## 2. Results

### 2.1. Dataset

A total of 5000 images were taken manually with cell phones during the growing season 2021/2022 at 14 sampling times from different angles and light conditions. The experiment was carried out with two genotypes, B481_6 and R453, with visual differences in their inflorescences. All the images were classified into five classes, as shown in Figure 1. Each class contained images of a particular developmental stage. The S1 class included all images of plants in the vegetative stages, regardless of the number of leaves. S2 included all reproductive stages described by Schneiter and J. F. Miller before the anthesis, referred to as the pre-anthesis period. S3 class is the anthesis stage and is characterized by the maturity of the ray flower, visible floret disks, stigma exposure, and dehisced anther locules. The S4 class included images belonging to the R6 reproductive stages referred to as the post-anthesis period, and no pictures of the R7 and R8 stages were taken. S5 class included images of plants at physiological maturity. 

### 2.2. Methods and Models 

Five different methods—CNN from scratch, VGG19, ResNet18, ResNet50, and VITB16—were evaluated using a balanced dataset of 450 images per class. In addition, data augmentation was implemented to augment the number of training samples artificially. Except for the CNN from scratch, transfer learning was applied.

A total of 300 epochs were performed for each method. Loss and accuracy were recorded in the validation and training sets for each epoch (Figure 2). During the training, the best model for each method was saved.

The best model for the CNN from scratch method is described in Table 1. Even so, VITB16 seems to have outperformed ResNet50 in the training set (Figure 2a,b), while, in the validation set, there are no clear differences among the Resnet18, ResNet50, VITB16, and VGG19 methods. The performance metrics of the testing set revealed notable differences among the evaluated models. The best-performing model within the ResNet50 approach exhibited remarkable accuracy (95.703%), coupled with a minimal error (0.043). Despite the ResNet50 model’s higher parameter count compared with ResNet18, it maintained a comparable prediction time, demonstrating its efficiency. VITB16 and ResNet18 models showed similar accuracy and error rates, and the performance of VGG19 and the CNN from scratch lagged, exhibiting the least favorable outcomes in terms of performance (Table 1). 

When evaluating the accuracy for individual classes, the model’s notable weakness lies in distinguishing between S3 and S4. This distinction is particularly evident in the CNN from scratch and VGG19 models. Interestingly, both ResNet18 and VITB16 outperformed ResNet50 in effectively distinguishing between these two stages (Figure 3).

### 2.3. Sunpheno 

The best model generated using the ResNet50 method was named Sunpheno. Sunpheno inferences in the test set showed that some pictures were incorrectly classified with high confidence (Appendix A). In this group of images, the largest portion belonged to the images of stage S3. 

It seems that the model has problems identifying the stages in pictures taken from the top when the floret weight turns down the plant (Figure 4a), and classification of pictures taken from the side (Figure 4b,c). Finally, one should consider that the prediction can be affected when there is overlap between vegetative and reproductive organs (Figure 4d).

### 2.4. Case Study

To test the Supheno model, we used it to monitor the developmental stage of sunflowers in two experiments carried out at INTA Castelar during the 2020/2021 and 2021/2022 growing seasons. The pictures were taken at 12 pm with variable weather conditions. The objective of these experiments was to compare the progress of senescence of genotypes B481_6 and R453 and its correlation with plant development. These genotypes have been widely studied because of their contrasting senescence phenotypes.

In both growing seasons (Figure 5a,b), the R453 genotype exhibited an early senescence phenotype in contrast with the B481-6 genotype. R453 showed half of the total amount of leaves senescent after 1100 °Cd and a positive increase in the slope after 1300 °Cd, reducing the plant lifespan. Meanwhile, genotype B481_6 showed half of the leaves senescent at 1600 °Cd with a diminished slope after 1300 °Cd. The obtained yield, measured as a weight of 1000 g, was significantly slower in the genotype R453 in both growing seasons, as was expected for this genotype (Figure 6a,b). 

Sunpheno successfully described the progression of developmental stages in both the growing seasons and genotypes. A total of 20 pictures for each sampling time and genotype were predicted (Figure 7). As time progressed, the proportion of predicted images shifted from S1 dominant at the beginning to S5 dominant at the later sampling date. No significant differences were found between genotypes, as most of the plants reached anthesis (S3) at approximately 850 °C and 950 °C. Before 168 °Cd in the growing season 21/22, Sunpheno had difficulty in assigning the label S1 to the images. This could be the result of the small size of the plants or the inference of mulch sheet.

## 3. Discussion

The detection of developmental stages, particularly in biological and agronomical contexts, often relies on analyzing various data points, such as images, genetic data, physiological measurements, etc. Implementing an intelligent agent (IA) for this purpose typically involves two key components, an expert eye on a field and a robust and well-supported machine-learning model [30]. A classic example is field phenotyping, which has unlocked new prospects for non-destructive field-based clean observation in plants for a large number of traits, including physiological, biotic (includes living factors such as fungi, bacteria, virus, insects, parasites, and weeds, etc.) and abiotic (includes non-living factors such as drought, flood, nutrient deficiency, senescence and other environmental factors) stress traits [31]. Determination of the phenological stages of plants is important for the growth of healthy and productive plants. The knowledge of transition times of phenological stages of a plant can provide valuable data for planning, organizing and timely execution of agricultural activities [32].

Due to the widespread availability of mobile phones and applications, tasks such as image classification can now be completed at the point of care. The deployment of biological image classification models on mobile phones can lead to several technical issues. For example, the model’s performance when trained or tested on a computer degrades significantly when the same model is deployed on a mobile phone [33].

The present study aimed to develop a tool for the automatic discrimination of sunflower phenological stages. Different deep machine-learning methods were applied to 2250 images, revealing that the ResNet50 architecture was the most accurate algorithm for this experiment, according to what happens in other systems, such as potato [34] and seldom crop systems [35].

The model generated in this study, Sunpheno, was used to evaluate changes in the developmental stages of the two genotypes with phenotypic differences in their inflorescences. These results are equivalent to those observed in previous studies [36]. R453 and B481-6 genotypes have similar phenological progressions, reaching anthesis at a similar time; however, senescence after anthesis in R453 was more pronounced than that in B481-6. This early senescence phenotype affected the observed yield of genotype R453.

Sunpheno successfully evaluated the changes in the developmental stages of the two genotypes, suggesting that it is robust enough to distinguish developmental stages, even if there are phenotypic differences between genotypes. However, additional genotypes and different growth treatments should be evaluated to confirm the application of this model.

The images in this work were taken with cell phones, which, unlike remote sensing tools, are laborious and have high dimensions as long as the image resolution level of a human is well trained for the physiological event addressed in the study, considering the elimination of the background as a key process for complex field trials and with the presence of the relevant biological information required to detect phenological stages [37,38]. However, evaluating developmental stages through a machine learning model homogenizes the results of these changes and removes subjectivity from the analyst measurements. As future work, hybrid techniques of machine vision and deep learning models can be applied in further field experiments to develop automated systems for precision agriculture based on cell phone images and a well-trained field observatory team. In addition, this model is intended as the first step in the development of a fully automatic monitoring experimental field, utilizing fixed cameras that can be installed on the ground or at the top of the antibird cages or even utilizing image acquisition from drones, as a future development for smart farming [39]. Additionally, CNN-based image analysis holds great promise for the accurate and efficient recognition of phenological stages in crops [40].

Finally, this study generated a 5000-image database, subdivided into 5 phenological stages that can be used in future computer vision studies. Hence, developing an IA machine learning model for detecting developmental stages is a multidisciplinary effort that combines data science and software engineering.

### Conclusions

In this study, a novel high-precision and high-efficiency method was proposed based on the deep learning convolutional neural network (CNN) model for the automatic detection of phenological stages in sunflower. Sunpheno has been shown to be a high-throughput and low-cost method, mainly developed and evaluated using images collected in a field, and can help breeders and farmers in decision making with regard to the early leaf senescence trait in cultivated sunflower.

## 4. Materials and Methods

### 4.1. Plant Material and Experimental Conditions

Two field experiments were conducted at the INTA Castelar Experimental Station (34°60′48″ S, 58°67′33″ W) during the 2020/2021 and 2021/2022 growing seasons.

Two sunflower inbred genotypes, R453 and B481-6, from the Instituto Nacional de Tecnología Agropecuaria (INTA) Sunflower Breeding Program, previously characterized as contrasting genotypes, were sown and evaluated for the senescence phenotype [36,41]. The plants were sown at 7.2 plants/m2 and cultivated under field conditions. The plants were grown under non-limiting water conditions, and soil water was maintained by irrigation. Diseases, weeds, insects, and birds were adequately controlled. Time was expressed on a thermal time basis by daily integration of air temperature with a threshold temperature of 6 °C and with plant emergence as the thermal time origin [42].

### 4.2. Physiological Parameters

Three plants were tagged, and the number of senescent leaves was calculated for each sample. After visual inspection, all leaves with more than 50% chlorotic tissue were considered senescent.

The phenological stages of the plants were followed every two days (growing season 2021/2022) or four days (growing season 2020/2021) by visual inspection. A stage was determined if more than 50% of the plants belonged to that stage.

When tagged plants reached physiological maturity, their heads were harvested. Seed number and seed weight per capitulum were also measured. Yield per genotype was calculated as the weight of 1000 seeds as follows: yield (g) = (FW(g) number of seeds) × 1000 [27].

### 4.3. Dataset

For each plant (~200 per genotype), pictures were taken during the life cycle on 14 dates during the season 2021/2022. Pictures were taken from different angles, distances and light conditions using three different smartphones. The resolution of the cameras was 14mpx and the image sizes varied between 4656 × 3492, 3016 × 4032, and 3024 × 4032. All of the images were resized to 224 × 224.

Twenty images from each genotype and date were separated, and the experimental set growing season was 21/22. The remaining pictures were classified into 5 different development stages: S1, S2, S3, S4, and S5, containing 450 images each. This dataset was random sized and split in three balanced sets: train, validation, and test in a relation 6.1.3.

During the season 2020/2022, 20 extra pictures were taken for each genotype on six dates along the life cycle of the plants (experimental set growing season 20/21).

### 4.4. Methods

We tested several methods for sunflower phenological stage classification from images, including convolutional neural networks and transformer vision, highlighting the memory requirements and the processing times to better compare the networks in the context of lightweight classification. 

#### 4.4.1. Baseline CNN

We designed a CNN from scratch to serve as the baseline. The network is composed of two convolutional layers followed by a fully connected layer. The two convolutional layers apply a series of four filters each of size 3 × 3. Stride and padding are set to 1, to preserve the input width and height. Batch normalization is applied after each convolutional layer, which normalizes the output of the convolutional layer to improve the stability and speed of training. Max pooling with 2 × 2 kernel is also applied in each layer, to help the network learn features that are more robust to input translations. A ReLU activation function is then applied elementwise to the output of the normalization layer, which introduces non-linearity and helps the network learn more complex representations. After the convolutional layers, a fully connected layer is applied to project the features to the final classification scores for the five classes (Appendix A). 

#### 4.4.2. VGG19

We chose to experiment with VGG19, a 19-layer convolutional neural network that belongs to the VGG family. This family of CNNs was introduced in [25] to perform classification on the ImageNet dataset. The authors sought explicitly to test the effect of the depth on the classification accuracy. The authors were also among the first to test very small (3 × 3) convolutional kernels in every layer. Thanks to its depth and its design, the VGG architecture achieved state of the art results in the ILSVRC-2014 competition. Specifically, VGG19 is composed of 16 convolutional layers followed by three fully connected layers. The concept is to reduce the spatial dimension of the feature maps, while increasing their number [25]. The reduction in spatial dimension is achieved using max pooling layers, while the increase in feature maps is achieved through the convolutional layers. The last three are a cascade of fully connected layers that project the features to the final output scores. Overall, VGG architectures have shown great accuracy using fewer parameters compared with counterparts such as AlexNet [25] (Appendix A). 

#### 4.4.3. ResNet Networks (18–50)

The family of ResNet architectures was introduced in [43]. At the time, researchers noticed that very deep networks were difficult to train, and that the deeper the network, the harder it was for it to converge. ResNet is the first work to introduce the concept of skip connections. Skip connections enable the direct transfer of information from one layer to another that is not adjacent in the network architecture. One of their main benefits is that they help to mitigate the vanishing gradient problem, which is a common issue in deep neural networks. The vanishing gradient problem occurs when the gradient of the loss function becomes very small as it propagates through the network during backpropagation, making it difficult to update the weights of the earlier layers. By using skip connections, the information from the earlier layers can be directly passed to the later layers, which can help to preserve the gradient and improve the flow of information through the network. Skip connections also allow for the creation of deeper and more complex neural networks without sacrificing performance. This is because the skip connections allow for the reuse of features learned in earlier layers, which can help to improve the overall accuracy and generalization of the network. Finally, skip connections can help to improve the speed of convergence during training. By allowing the network to access information from earlier layers, skip connections can help to speed up the learning process and reduce the number of iterations required to reach convergence. We will experiment with two ResNet variants, resnet18 and resnet50. The main difference between them is the depth of the network, one is characterized by 18 layers, while the second by 50 layers. Additionally, resnet18 and resnet50 employ two different building blocks. Resnet18, given the contained number of layers, adopts a basic block formed by two 3 × 3 convolutions that reduces the spatial dimension by increasing the number of feature maps. In contrast, resnet50 adopts a basic block formed by three convolutions in series. The first exploits a 1 × 1 kernel to reduce the number of feature maps to improve the efficiency of the 3 × 3 convolution that follows. Finally, another 1 × 1 convolution brings the number of feature maps back to the initial one. This process allows the network capacity to be maintained by using fewer parameters, especially in very deep networks. In fact, this approach is also used in other ResNet variants, such as resnet-101 or resnet-152 (Appendix A).

#### 4.4.4. Transformer

The Transformer architecture [44], originally developed for natural language processing (NLP) tasks, has recently gained attention in image processing as an alternative to convolutional neural networks (CNNs). CNNs have been highly successful in image processing tasks, but they struggle with capturing long-range dependencies due to the limited receptive field of the convolutional kernel. The Transformer architecture, on the other hand, relies solely on self-attention mechanisms and does not rely on sequential processing. This allows it to capture dependencies between distant elements in the input sequence without being limited by the receptive field of the convolutional kernel. Moreover, the self-attention mechanism of the Transformer architecture enables it to attend to different parts of the input image, making it well-suited for image processing tasks that require the capturing of relationships between non-adjacent pixels. Recent studies have shown that the Transformer architecture can achieve competitive results on several tasks, such as image classification, object detection, and segmentation. Its efficient use of memory and its ability to handle long-range dependencies make it a particularly appealing choice for these tasks, especially in cases where the input image has complex and non-local dependencies. While CNNs remain the dominant approach for image processing tasks, the Transformer architecture presents a promising alternative, with its ability to capture non-local dependencies and attend to different parts of the input image; as a result, we decided to include it in our analysis (Appendix A). 

### 4.5. Transfer Learning

Transfer learning is a machine-learning technique in which a model trained on one task is used as a starting point for a different but related task. The so-called “pre-trained” model is then used to extract general features that can be used as input for a new model trained on a specific task. In the computer vision community, models pre-trained on the ImageNet [24] dataset have been a longstanding starting point for a series of downstream tasks, including image classification.

One of the key benefits of transfer learning is that it can significantly reduce the amount of data required to train a new model. By using a pre-trained model as a starting point, the knowledge already captured in the pre-trained weights can be leveraged to improve performance and achieve a faster convergence. Additionally, transfer learning can help mitigate the problem of overfitting, as the pre-trained model has already learned general features that can be used to regularize the new model. In the context of image classification, transfer learning is usually achieved by removing the last classification head and replacing it with a different head to accommodate for the number of output classes. The network is then trained by freezing the parameters of all but the last classification head. This prevents the pretrained weights from changing during training and allows the classification head to adapt to the given task and the extracted features. 

### 4.6. Data Augmentation

Data augmentation is a technique used to artificially increase the size of a dataset by creating new, modified versions of existing data. With classification, several different transformations can be applied to augment the dataset. In this work we applied transformation to the input images to increase the dataset variability. Randomly, 20% of the input images are rotated either in 90, 180, or 270 degrees; 20% are changed by increasing or decreasing the brightness; and 20% are horizontally or vertically flipped.

### 4.7. Experimental Operation Environment

The proposed model was implemented in Python using the PyTorch library and trained with an NVIDIA Quadro P2000 GPU (5 GB). The initial learning rate is 0.007, which varies in a Poly manner. The maximum number of epochs used for training is 300, while the batch size is 8. We used an Adam optimizer with a weight decay factor of 0.0001 on the parameters to prevent overfitting.

### 4.8. Performance Evaluation

Precision, recall, and macro F1 score are important metrics used to evaluate the performance of classification models. Precision (Pr) measures the proportion of correctly predicted positive instances among all instances predicted as positive, highlighting the model’s accuracy in identifying true positives (TP) while minimizing false positives (FP). Recall (Rc), on the other hand, assesses the ratio of correctly predicted positive instances to the total actual positive instances, focusing on the model’s ability to capture all positive instances and reducing false negatives. Macro F1 score combines both precision and recall, providing a balanced measure of a model’s effectiveness by considering their harmonic mean. It is particularly useful when dealing with imbalanced classes, as it gives equal weight to each class and offers a comprehensive assessment of a model’s performance across all classes. The overall performance of each method was evaluated with the following metrics, as follows:$$Precision=Pr_{i}=\frac{TP_{i}}{TP_{i}+FP_{i}}$$
$$Recall=Rc_{i}=\frac{TP_{i}}{TP_{i}+FN_{i}}$$
$$Marco\;Average\;Precision=MPr=\frac{\sum_{i}Pr_{i}}{N}$$
$$Marco\;Average\;Recall=MRc=\frac{\sum_{i}Rc_{i}}{N}$$
$$Marco\;F1\;score=2\times \frac{MRc\times MPr}{MRc+MPr}$$
Error=1−MRc
$$Confidence\;interval=1.96\times \sqrt[{2}]{\frac{MRc\times Error}{Ns}}$$
Accuracy=Marco Average Recall×100
Accuracy±=Confidence interval×100
where *i* denotes the class (S1, S2, S3, S4, S5).

## Figures and Tables

**Figure 1 plants-13-01998-f001:**
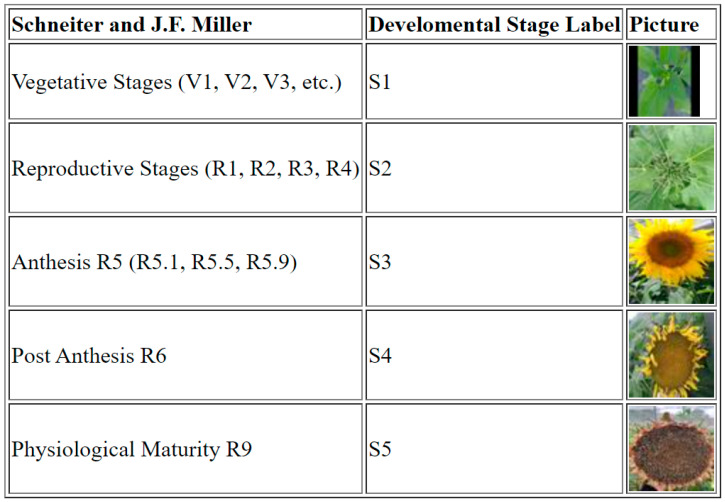
Image dataset classified into five classes based on Schneiter and J. F. Miller phenological stages description.

**Figure 2 plants-13-01998-f002:**
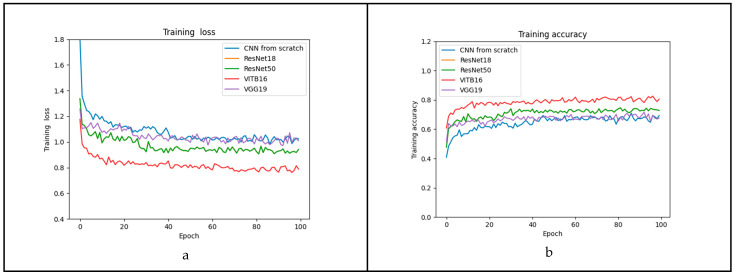
Loss and accuracy during training were evaluated in the training and validation sets in each epoch. (**a**) Training loss, (**b**) training accuracy, (**c**) validation loss, and (**d**) validation accuracy. The first 100 epochs are shown.

**Figure 3 plants-13-01998-f003:**
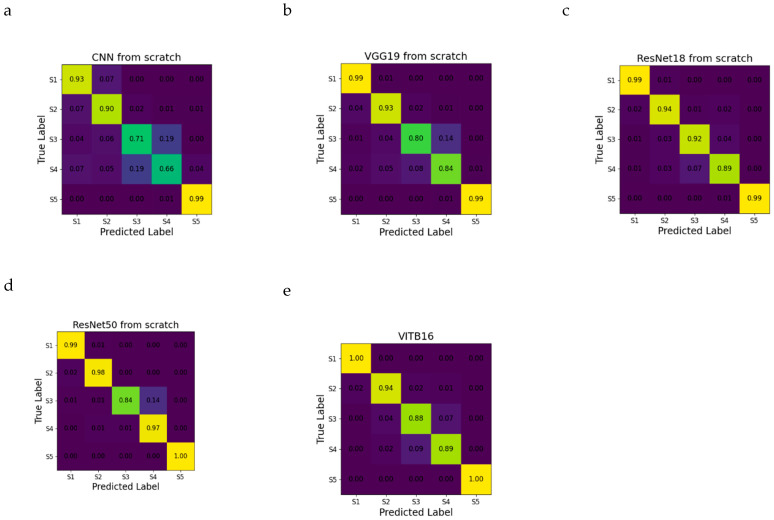
Confusion matrices for the sunflower datasets obtained in the test set. (**a**) CNN from scratch, (**b**) VGG19, (**c**) ResNet18, (**d**) ResNet50, and (**e**) VITB16. For each matrix, the rows show the true development stage, whereas the columns show the predictions from the CNN architecture. The accuracy of the prediction of each class is shown. Accuracy values from 0 to 1 are represented from dark purple to light yellow.

**Figure 4 plants-13-01998-f004:**
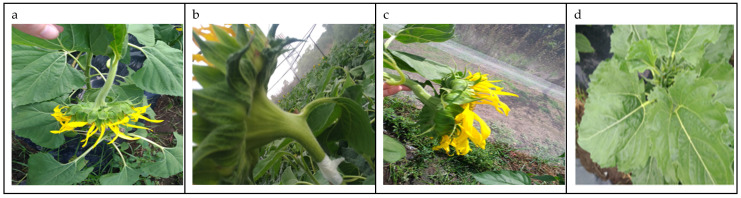
Examples of images incorrectly classified by the best model, ResNet50, with high confidence in the prediction. (**a**) Image of phenological stage S3, incorrectly classified as S4 with a confidence of 0.74. (**b**) Image of phenological stage S4, incorrectly classified as S2 with a confidence of 0.77. (**c**) Image of phenological stage S3, incorrectly classified as S4 with a confidence of 0.79. (**d**) Image of phenological stage S2, incorrectly classified as S1, with a confidence of 0.65.

**Figure 5 plants-13-01998-f005:**
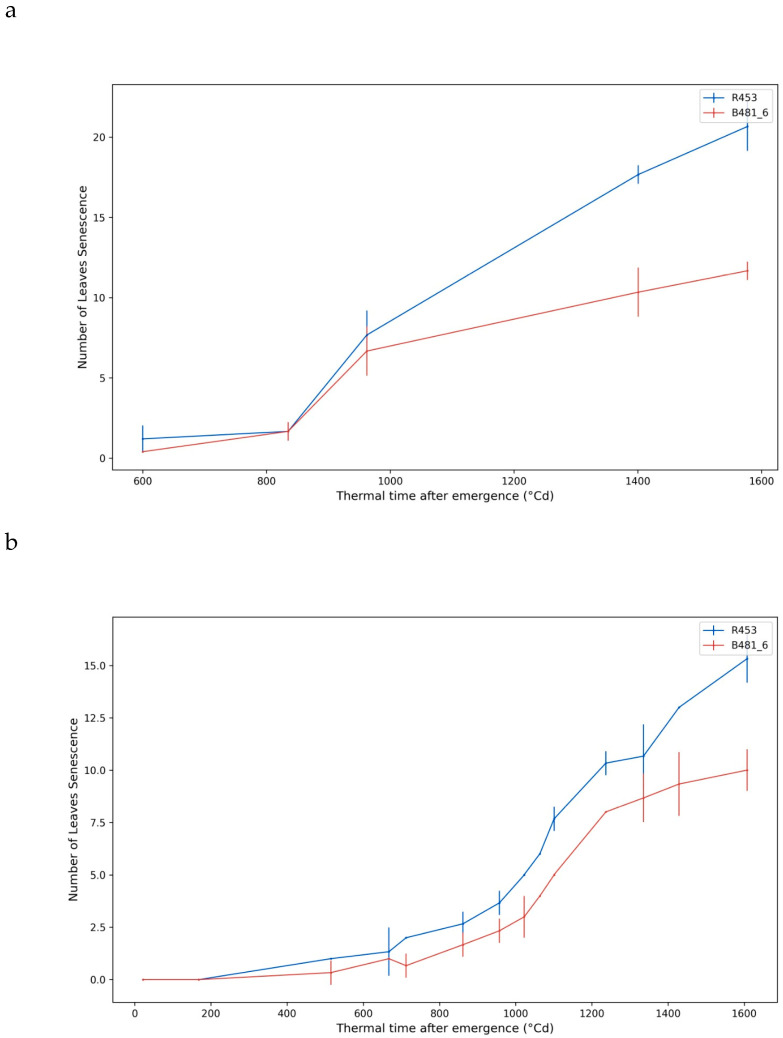
Number of senescent leaves over time. (**a**) Growing Season 20/21, with the measurements taken on five sampling dates. (**b**) Growing Season 21/22, with the measurements taken on fourteen sampling dates. The time was recorded as the thermal time after emergence (°Cd). The red and blue lines represent the genotypes B481_6 and R453, respectively.

**Figure 6 plants-13-01998-f006:**
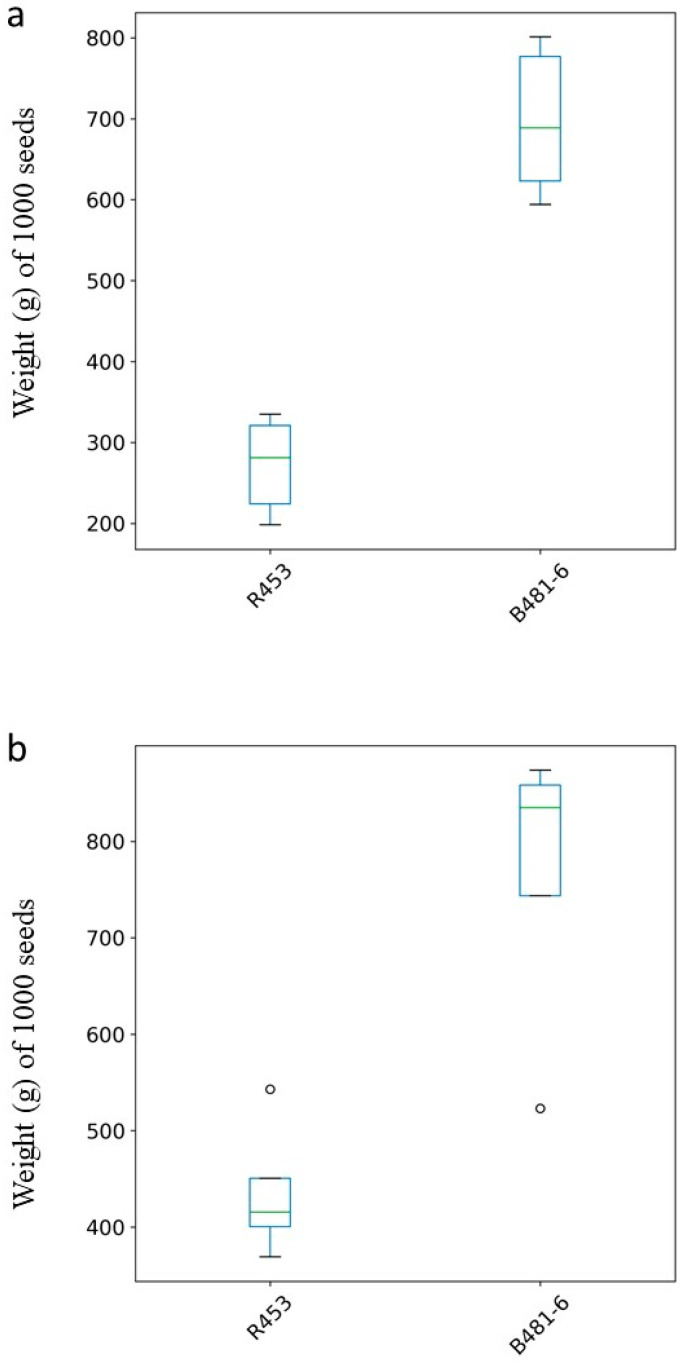
Box plot showing the weight in grams of 1000 seeds for R453 and B481_6 genotypes. (**a**) Growing Season 20/21 (**b**) Growing Season 21/22. Outliers are shown as open circles outside the boxes. The top and bottom of the box represent the first quartile (Q1) and third quartile (Q3), respectively. This range is known as the interquartile range (IQR) and contains the middle 50% of the data. A green line inside the box represents the median (Q2) of the data.

**Figure 7 plants-13-01998-f007:**
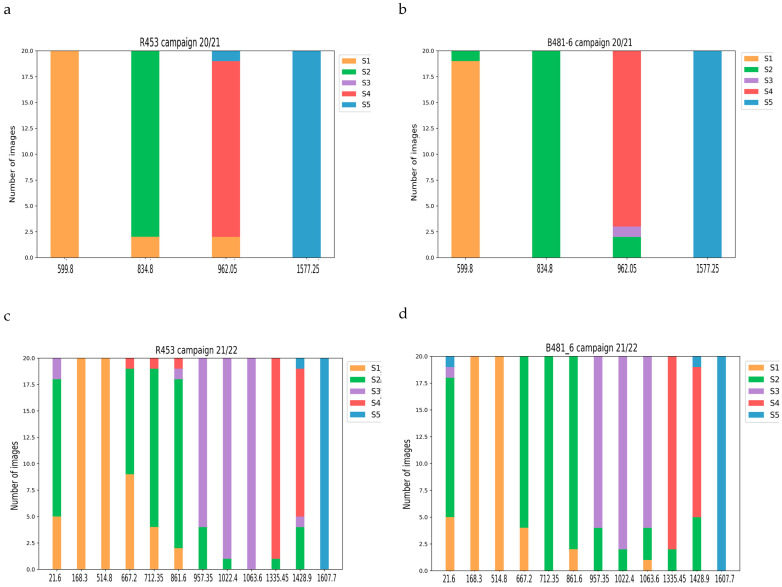
Sunflower stages predicted with Sunpheno. (**a**) Genotype R453, growing season 20/21. (**b**) Genotype R481-6, growing season 20/21. (**c**) Genotype R453, growing season 21/22. (**d**) Genotype R481-6, growing season 21/22. The time was recorded as the thermal time after emergence (°Cd). For each time, 20 images were predicted. Yellow bars indicate stage S1, green S2, purple S3, red S4 and blue S5.

**Table 1 plants-13-01998-t001:** Performance metrics.

	CNN from Scratch	VGG19	ResNet18	ResNet50	VITB16
Macro average precision	0.83	0.91	0.95	0.96	0.94
Macro average recall	0.84	0.91	0.95	0.96	0.94
Macro F1 score	0.84	0.91	0.95	0.96	0.94
Error	0.16	0.09	0.05	0.04	0.06
Confidence interval (95%, z = 1.96)	0.03	0.02	0.02	0.02	0.02
Accuracy	83.70	90.96	94.52	95.70	94.22
Accuracy +/−	2.79	2.16	1.72	1.53	1.76
Number of parameters	63,001	139,601,733	11,179,077	23,518,277	85,802,501
Time of test 675 images (s)	**63,091**	**18,552**	**17,252**	**18,542**	**18,049**

## Data Availability

For making inferences in images a google share folder was created (https://drive.google.com/drive/folders/1q39dsbNmvoh8t_jSCSsprHNWZw78yYYV?usp=drive_link (accessed on 24 June 2024)). The subfolder Using_the_model contains Google collabs (or Python notebook) that can be used as a tutorial. The Google collabs connects to the Sunpheno model in order to make inferences about the images save on the subfolder ‘pics’. After running the notebook, the output table with the name of the images, inference value, and confidence in the prediction are saved in the subfolder ‘results’. The whole project containing the scrips used are available in the same google share folder ‘Sunflower_Stages’. Image dataset supporting the findings of this study are available upon reasonable request, due to the storage size required. A public repository hosting the source codes and a sample subset of the dataset used in the current work will be made promptly available at INTA hosting.

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
