# Peer review of "Sunpheno: A Deep Neural Network for Phenological Classification of Sunflower Images"

_plants, 2024, doi:10.3390/plants13141998_

Round 1

Reviewer 1 Report (Previous Reviewer 1)

Comments and Suggestions for Authors

The manuscript title “Sunpheno: a deep neural network for phenological classification of sunflower images” is a good study and have scientific worth. This MS need minor improvements especially in the results section.

Summary of this study:

Leaf senescence is crucial for grain filling in plants, requiring synchronization with phenological stages for optimal yields. This study evaluated five deep machine-learning methods for assessing sunflower phenological stages using cell phone images in the field. The method based on the pre-trained network resnet50 showed superior performance in accuracy and speed compared to other methods. The developed model, Sunpheno, successfully evaluated phenological stages of sunflower lines B481_6 and R453 during senescence, revealing clear differences between the lines. A database of 5,000 images classified by an expert was created to reduce subjectivity in field measurements, correlating phenological stages with performance and senescence parameters crucial for yield improvement.

Reviewer Comments:

1-      Don’t split the introduction section in to several paragraphs such as combine the line 75-76 paragraphs and lines 91 and 93.

2-      In the table 2: Number of Parameters 63,001 139,601,733 11,179,077 23,518,277 85,802,501. Why they are so large numbers?

3-      In table 2: please add the software name that authors have used to calculate these values, in the table legends.

4-      Also correct the punctuation and other mistakes in the manuscript such as table 2 last tine:  (TIme of test), line 12: yields.. and so on…

5-      Figure 3 and in all figures: the figure legends must be shown below the figure.

6-      Figure 5: what does the graphs bars means? Either it is standard error or devia.

7-      Have authors used statistical software to check the level of significance in figure 5?

8-      Figure 6 captions front are very small, hard to read, if possible than do a slightly large front.

9-      Please add a conclusion section of this study. 

Comments on the Quality of English Language

English is mostly fine, some punctuation issues.

Author Response

Reviewer 1 Comments:

1-      Don’t split the introduction section in to several paragraphs such as combine the line 75-76 paragraphs and lines 91 and 93.

Done

2-      In the table 2: Number of Parameters 63,001 139,601,733 11,179,077 23,518,277 85,802,501. Why they are so large numbers?

 The large numbers of parameters in Table 2 are due to the nature and complexity of the machine learning models we have used.

3-      In table 2: please add the software name that authors have used to calculate these values, in the table legends.

All the routines were run in Python (www.python.org). Any software was used.

4-      Also correct the punctuation and other mistakes in the manuscript such as table 2 last tine:  (TIme of test), line 12: yields.. and so on…

Corrected.

5-      Figure 3 and in all figures: the figure legends must be shown below the figure.

Done.l

6-      Figure 5: what does the graphs bars means? Either it is standard error or devia.

This is a boxplot. A boxplot, also known as a box-and-whisker plot, is a standardized way of displaying the distribution of data based on a five-number summary: minimum, first quartile (Q1), median, third quartile (Q3), and maximum. Here are the key components of a boxplot, including the "bars":

Box: The box itself represents the interquartile range (IQR), which is the range between the first quartile (Q1) and the third quartile (Q3). This range contains the middle 50% of the data.

  • Bottom of the Box: This line indicates the first quartile (Q1), which is the median of the lower half of the dataset.
  • Top of the Box: This line indicates the third quartile (Q3), which is the median of the upper half of the dataset.

Median Line: Inside the box, there is a line that represents the median (Q2) of the dataset. This is the midpoint of the data, where 50% of the values are below it and 50% are above it.

Whiskers: These are the "bars" that extend from the box. They indicate the variability outside the upper and lower quartiles.

  • Lower Whisker: Extends from Q1 to the smallest value within 1.5 * IQR of the lower quartile.
  • Upper Whisker: Extends from Q3 to the largest value within 1.5 * IQR of the upper quartile.

Outliers: Data points outside the range of the whiskers are considered outliers and are often plotted as individual points.

7-      Have authors used statistical software to check the level of significance in figure 5?

This was a Python output which has a strong level of confidence due to the significant results obtained for this graph in yield measured as a weight of 1000 g.

8-      Figure 6 captions front are very small, hard to read, if possible than do a slightly large front.

Done

 9-      Please add a conclusion section of this study. 

Done.

Reviewer 2 Report (Previous Reviewer 2)

Comments and Suggestions for Authors

The responses of the authors to the reviewer’s comments appear to be reasonable. However, this work takes a very big number of images manually, and thus is much more time-consuming, laborious, expensive, and costly than the professional systems, such as Unmanned Aerial Systems (UAS).

The authors have made very big efforts on this system. I believe that the future, if any, of this system, will rely on taking images automatically, not manually.

I still don’t understand many sentences and words, such as the following:

Line 21to 22, “This is relevant to finish with subjetivity between non-trained people measuring this trait in the field”

Line 54, “by”

Lines 261 to 262, “…as long as a well-trained human level of image resolution for a complex trait, considering back ground removal, relevant for complex field assays, but with relevant biological information when detecting phenological stages, among others.” 

Comments on the Quality of English Language

Many sentences and words have problems. I suggest that authors ask English editing services or an expert for help on English. 

Author Response

Reviewer 2 Comments:

The responses of the authors to the reviewer’s comments appear to be reasonable. However, this work takes a very big number of images manually, and thus is much more time-consuming, laborious, expensive, and costly than the professional systems, such as Unmanned Aerial Systems (UAS).

The authors have made very big efforts on this system. I believe that the future, if any, of this system, will rely on taking images automatically, not manually.

 Thank you very much for your suggestion. We are exploring this manual tool in order to detect sunflower developmental stages within a frame of a highly accurate field assay due to the fact that any information is available for this oil relevant crop. Once the algorithm is fully trained for the senescence process at the field, our idea is to process them automatically based on drone images.

I still don’t understand many sentences and words, such as the following:

Line 21to 22, “This is relevant to finish with subjetivity between non-trained people measuring this trait in the field”

The sentence was rewritten.

Line 54, “by”

The sentence was rewritten.

Lines 261 to 262, “…as long as a well-trained human level of image resolution for a complex trait, considering back ground removal, relevant for complex field assays, but with relevant biological information when detecting phenological stages, among others.” 

The sentence was rewritten.

Many sentences and words have problems. I suggest that authors ask English editing services or an expert for help on English. 

The manuscript was fully revised and edited. Thank you very much for your suggestion.

This manuscript is a resubmission of an earlier submission. The following is a list of the peer review reports and author responses from that submission.

Round 1

Reviewer 1 Report

Comments and Suggestions for Authors

The article “Sunpheno: a deep neural network for phenological classification of sunflower images” has scientific worth, but has major defaults and requires significant revisions. I suggest authors rewrite the discussion section/ adjust the manuscript (result section), methods, and reference section.

Summary of current manuscript:

Leaf senescence is a crucial stage in leaf development, marked by a decline in photosynthetic rate, nutrient recycling, and cell death, impacting crop yield. Genotypes with early senescence are associated with lower yield and seed quality. Traditional phenology studies are labor-intensive, but high-throughput phenotyping techniques, including machine learning methods, offer efficient alternatives. The current study evaluated five deep machine-learning methods for assessing sunflower phenological stages using cell phone images in the field. The method based on the pre-trained network resnet50, named Sunpheno, outperformed others in accuracy and speed. Sunpheno successfully evaluated phenological stages of sunflower lines B481_6 and R453 during senescence, showing clear differences in stages, aligning with previous studies.

My comments for authors are as follows:

1-      In abstract the background is too large Line 10-21: I suggest author’s to minimize it in 3-4 lines. Please rewrite the abstract, mostly describing the results… such as 2-4 lines background, 1-2 lines scientific reasons/importance of this research, 1-2 methods, 5-8 results, 2-3 concluding lines.

2-      Line 33: Don’t use “,” to separate the decimals (21,4). Write like this “value of 21.4 billion USD”

3-      Line 40, 43, 57 and so on: Use recommended citation styles according to Plants journal.

4-      Line 112-114: Please delete these lines. “This section may be divided by subheadings. It should provide a concise and precise description of the experimental results, their interpretation, as well as the experimental conclusions that can be drawn.”

5-      Line 116: Same issue as comment 2. “A total of 5,000 images”

6-      Please mention the time (AM/PM) and environmental condition in which the pictures are taken.

7-      Line 115: “2.1.1 Dataset” why 2.1.1? it should be 2.1.

8-      Please move the table 1 above, where it was cited in the text, in 2.1 Dataset heading. Also move the table 2 above where it was cited in the text.

9-      If possible please explain the line 137-142 results in a small table form, it would be easy to understand.

10-   Please explain what is this “Number of Parameters 63,001 139,601,733 11,179,077 23,518,277 85,802,501” in table 2?

11-   Figures 4 and 5 there is no significant difference? What is meant by bars in both figures? Is St. error or St. dev.?? Explain the each abbreviation of figures and table in their respective legends.

12-   All the figures and tables should be placed, where they was cited first time in the text.

13-   Line 272-275: delete these lines. Why authors keep these lines?

14-   Line 276: Discuss your results and defend them with previously published manuscripts don’t explain aim of this study in the discussion section.

15-   Line 276-300: Discussion section is too small and without references??? It’s first time I am reviewing a discussion section without references.

16-   Discussion section: Line 276-279: explain the aim of this study, Line 280-290: Explain the results of this study. Line 291 and 299 of discussion section, already described in line 116 results section.

17-   Line 308: Reference [75] was cited, and in line 313: Reference [77] was cited about in manuscript, however, overall this manuscript have 29 references???????? 

Comments on the Quality of English Language

Some English paragraph needs rhythm.

Reviewer 2 Report

Comments and Suggestions for Authors

High-throughput phenotyping is much needed in crop phenotyping area. The authors used smart phones as an image tool for high-throughput phenotyping, and intended to make this system as an alternative to other platforms. The idea appears to be innovative, materials, image acquisition, data processing and results appear to be reasonable. The English writing is readable except for some minor typos, grammatical errors and a few minor writing problems. However, this work suffers from several major problems, including

(1) Before this system is accepted for scientific research work in plant phenotyping community, this system may be required to be compared (validated) with professional high-throughput phenotyping platforms, e.g., UAS and RGB systems.

(2) Smartphone cameras are designed for images for human eyes. Plants absorb and reject different light wave lengths when compared to human eyes.

(3) Implementing this system in the scientific community may be misleading.

Some minor typos, grammatical errors and writing problems including

(1) Line 33 to 36

(2) Line 44, during

(3) Line 101, in

Comments on the Quality of English Language

Readable.